

# Validation of the HeLD-14 functional oral health literacy instrument in a general population

Priscilla Flynn[1], Aparna Ingleshwar[2], Xing Chen[3], Leah Feuerstahler[3], Yvette Reibel[4] and Mike T. John[2]

[1] Primary Dental Care, School of Dentistry, University of Minnesota, Minneapolis, MN, United States of America
[2] Diagnostic and Biological Sciences, School of Dentistry, University of Minnesota, Minneapolis, MN, United States of America
[3] Department of Psychology, Fordham University, Bronx, NY, United States of America
[4] Primary Dental Care, School of Dentistry, University of Minnesota, Minneapolis, MN, United States of America

Corresponding author
Priscilla Flynn, flynn125@umn.edu

## ABSTRACT

**Background.** Oral health literacy (OHL) is recognized as an important determinant of oral outcomes. Measuring OHL with a valid and reliable instrument that accurately captures the functional nature of this construct across cultures is needed. The short version of the Health Literacy in Dentistry scale (HeLD-14) shows promise as an appropriate instrument due to its inclusion of comprehensive domains hypothesized to comprise OHL. While studies validating the instrument in several languages have occurred, the number of dimensions in the various analyses range from one to seven. Validation of the HeLD-14 in a general English-speaking population is also lacking. The purpose of this study was to explore and confirm the dimensionality of the HeLD-14 in a general US English-speaking population.

**Methods.** The psychometric properties of HeLD-14 were evaluated in a sample of 631 participants attending the Minnesota State Fair. Construct validity was assessed using exploratory factor analysis (EFA) followed by confirmatory factor analysis (CFA) on the data set split into two groups. Internal consistency reliability was assessed using the Cronbach's alpha coefficient. Concurrent validity was established between the HeLD-14 and the Oral Health Inventory Profile (OHIP-5) using Pearson's correlation.

**Results.** EFA found, and CFA reinforced, a unidimensional structure of the HeLD-14. Cronbach's alpha was acceptable at 0.92. Fit assessment also supported a unidimensional structure, comparative fit index = 0.992, Tucker-Lewis index = 0.991, root mean square error of approximation = 0.065, and standardized root mean square residual = 0.074. Concurrent validity analyses showed that the HeLD-14 correlated with the OHIP-5.

**Conclusions.** The HeLD-14 is a unidimensional reliable and valid instrument for measuring the oral health literacy in the general US English-speaking adult population.

## INTRODUCTION

Oral health literacy (OHL) is defined as, "the degree to which individuals have the capacity to obtain, process, and understand basic oral health information and services needed to make appropriate health decisions and act on them" (*US Department of Health Human Services, 2000*). Estimates of OHL prevalence are equated with general health literacy, estimating that only one in ten adults fully understand written health material (*Institute of Medicine Committee on Health Literacy, 2004*). Individuals with low general health literacy are more likely to have low OHL and are less likely to utilize preventive care, have poorer overall and oral health, and higher mortality rates compared to those with adequate health literacy (*Baskaradoss, 2018*; *Kleinman, Horowitz & Atchison, 2021*; *National Institutes of Health, 2021*). The recognition of the impact of OHL on health outcomes is reflected in recent national initiatives to measure oral health literacy. Health literacy is a central focus of Healthy People 2030 (*Office of Disease Prevention and Health Promotion, 2021*). One of the initiative's overarching goals is to, "Eliminate health disparities, achieve health equity, and attain health literacy to improve the health and well-being of all" (*Office of Disease Prevention and Health Promotion, 2021*). Similarly, the Strategic Research Plan of the National Institute of Dental and Craniofacial Research includes the objective to, "Accelerate research on best practices of ... health literacy that overcome obstacles and improve behaviors and practices" (*National Center for Dental Craniofacial Research, 2021*). Evaluating these proposed improvements in OHL among the American population requires designing and testing valid measurement instruments.

Researchers have developed several OHL instruments, the majority of which are based on two proposed constructs: reading and numeracy (*Dickson-Swift et al., 2014*). Subsequent instruments proposed additional constructs hypothesized to better characterize functional OHL, consisting of four to seven constructs. One such instrument is the 29-item Health Literacy in Dentistry (HeLD-29) scale, initially designed to measure OHL in an Australian Aboriginal population (*Jones et al., 2014*). Adapted from the Health Literacy Measurement Scale (HeLMS) (*Jordan, Buchbinder & Osborne, 2009*), seven constructs were hypothesized described as representing receptivity, understanding, support, economic barriers, access, communication, and utilization (*Jones et al., 2014*). Results of factor analysis found a unidimensional solution following exploratory factor analysis (EFA). The authors reported low, yet statistically significant, correlations between the seven latent variables, leading to confusion about the dimensionality of OHL (*Jones et al., 2014*). A shortened version, the HeLD-14 consisted of two items for each of the seven constructs, and was designed for settings where only a limited number of items can be administered (*Jones et al., 2015*). Factor analysis of the shortened version found six rather than seven of the original principal components as "utilization" dispersed onto the "communication" domain (*Jones et al., 2015*). A Chinese version (HeLD-C) found a unidimensional solution and validated the instrument in an adult Chinese population (*Lui et al., 2021*). Two subsequent studies both found six dimensions in a small sample of Thai adults with special needs, and a large population of Brazilian non-institutionalized older adults (*Sermsuti-Anuwat & Piyakhunakorn, 2021*; *Soares et al., 2022*). While the

dimensionality of the HeLD remains undetermined, its' comprehensive foundation makes it an instrument of interest in measuring functional oral health literacy. Discrepancies in dimensionality are not unusual when analyzing newer instruments. Previous analysis of another functional OHL instrument, the Oral Health Literacy-Adult Questionnaire (OHL-AQ), was found to be unidimensional despite prior publications suggesting multiple dimensions (*Flynn, John & Sistani, 2018*; *Sabbahi, Limback & Rootman, 2009*; *Sistani et al., 2013*). In addition to identifying construct dimensionality, studies intended to reduce the number of items in a validated instrument is standard practice to limit survey burden for participants, particularly when administering a survey in a clinical or community setting. One instrument of note is the Oral Health Impact Profile (OHIP) designed to measure Oral Health-Related Quality of Life (OHRQoL). The original version consisted of 49 items and was subsequently reduced to 14 items that increased its utility (*Slade & Spender, 1994*). The most recent version consists of five items and has been shown to be valid and reliable in multiple languages (*John et al., 2014*; *John et al., 2006*; *Simancas-Pallares et al., 2020*). In order to assure the validity and reliability of the HeLD-14, the authors intend to follow the research processes used for the OHIP-5 and other similar instruments.

The purpose of this study, therefore, was to investigate the dimensionality of and validate the HeLD-14 in a general U.S. population. The study's research hypothesis was that the HeLD-14 is unidimensional.

## MATERIALS & METHODS

Ethical approval. The study was approved by the University of Minnesota Institutional Review Board (STUDY00016028).

### Study setting and participants

The study was conducted at the Minnesota State Fair, the second largest in the United States, and two county fairs in Southern Minnesota. As fairs attract a wide range of individuals, the setting provides a unique environment to administer and validate surveys in a general population rather than a clinical or controlled laboratory setting. Study eligibility included adults aged 18 and older whose cognitive and English-language capabilities were adequate to discuss the purpose of the study and complete the survey. Recruitment consisted of discussing the study's purpose with potential participants approaching the study booth. Interested individuals were given an electronic tablet to complete a written consent form followed by the survey. Upon survey completion, participants were given a brief intraoral clinical assessment. All participants were required to be able to have adequate cognitive, auditory and visual abilities to converse verbally with the study staff, complete the survey on the electronic tablet, and hold their mouth open for several minutes to complete the oral assessment. Participants who were unfamiliar with electronic tablet use were assisted by study staff.

### Survey instrument

The survey consisted of the HeLD-14, the Oral Health Inventory Profile (OHIP-5) and demographic factors. Each HeLD-14 item is prefaced with the stem ''are you able to…''
or "do you know...". Two items from each of the seven proposed OHL domains are summarized as follows: communication (look for a second opinion, use information), understanding (read written dental information, read dental information brochures), receptivity (make time for dental health needs, pay attention to things good for dental health), utilization (carry out dental instructions, use dentist advice), support (take support to a dental appointment, ask for support for a dental appointment), financial (pay to see a dentist, pay for dental medication) and access (how to get a dental appointment, what to do to get a dental appointment). For each item the focus was on 'difficulty' experienced. Five-point adjectival scale choices ranged from "unable to do" = 0 to "no difficulty" = 4 such that responses in higher categories indicated greater oral health literacy (*Jones et al., 2014*).

The HeLD-14 was compared to the OHIP-5 to establish concurrent validity. The OHIP-5 is a valid and reliable instrument that assesses OHRQoL. A previous study confirmed the predictive ability of the initial HeLD instrument with a comparison to the OHIP-14 (*Bado et al., 2020*). Our study used the OHIP-5 to replicate the findings of the aforementioned study, but using the short form versions of both instruments.

## Analysis

Descriptive statistics were calculated for demographic factors reporting means and standard deviations for continuous variables, and counts and proportions for categorical variables.

Because the seven-dimensional structure of the HeLD has not been consistently replicated (*Jones et al., 2015*; *Lui et al., 2021*; *Soares et al., 2022*; *Bado et al., 2020*), conducting exploratory factor analysis was considered the most appropriate and conservative approach. Therefore, to investigate the factor structure and validate these findings, we randomly split the data into an exploratory ($N = 316$) and a confirmatory ($N = 315$) dataset to use independent data for each research step. To assure adequate sample sizes for factor analysis, lower sample sizes are required if factor loadings are high and there exist more items per factor. Previous research has demonstrated that samples as low as 100–200 are sufficient if there are at least four items per factor and loadings are very high (*e.g.*, 0.7 or higher) (*Macallum et al., 2001*; *MacCallum et al., 1999*). These factor loadings are similar to those found in previous studies of the HeLD-14 (*Jones et al., 2014*), and therefore our sample sizes of >300 for both the EFA and CFA are sufficient.

Exploratory factor analysis was conducted based on the polychoric correlation matrix of the 14 items. Polychoric correlations were used to account for the ordinal nature of responses to the HeLD-14. To determine the number of factors, a scree plot was generated followed by finding the ratio of first to second eigenvalues, and parallel analysis was completed. The scree plot graphs the eigenvalues in decreasing order and retains as many factors as there are eigenvalues above the elbow of the plot. Horn's parallel analysis modifies this scree plot by comparing the observed eigenvalues to random simulated eigenvalues, retaining as many factors as the number of observed eigenvalues that exceed the simulated eigenvalues. For the ratio of first-to-second eigenvalues, a ratio larger than 4 can be interpreted as evidence of unidimensionality (*Reeve et al., 2007*). After deciding on

the number of factors, an EFA was conducted using the unweighted least squares (ULS) estimation which does not make distributional assumptions about the data.

Next, confirmatory factor analysis (CFA) was conducted to validate the structure found in the EFA. Because responses to this assessment are on a five-point scale, an ordinal CFA model was fit using ULS estimation. Model fit was evaluated using $\chi^2$ (df) and the corresponding $p$ value, the root mean square error of approximation (RMSEA), standard root mean squared residual (SRMR), comparative fit index (CFI) and Tucker-Lewis index (TLI). According to previous research, acceptable model fit guidelines include a non-significant $\chi^2$ test, RMSEA $\leq$.06, SRMR $\leq$ 0.08 and TLI and CFI $\geq$0.95 (*Hu & Bentler, 1999*).

To measure concurrent validity, HeLD-14 sum scores were correlated with the OHIP-5 sum scores.

The analysis was completed using Stata/SE version 17.0 software (*StataCorp, 2021*).

## RESULTS

Surveys were initiated by 651 participants at two Minnesota County ($n = 63$) and the State Fair ($n = 588$) with 631 completed surveys available for analysis. Characteristics of the study population are reported in Table 1. The majority of participants were female, White, non-Hispanic, reported completing more than four years of college, and the average age was 43.6 years.

The mean response and interquartile range (IQR) of each HeLD-14 item are reported in Table 2. All items exhibited a mean score close to the maximum score of five and a small IQR of zero or one, reflecting that most members of the surveyed population indicate high oral health literacy.

### EFA results

In the exploratory dataset, we used a scree plot, parallel analysis, and the ratio of the first to second eigenvalue to determine the number of factors. In Fig. 1, the scree plot indicates that one factor accounts for most of the variance in the exploratory dataset. Parallel analysis also indicated a one factor solution, and the ratio of first to second observed eigenvalue was 12.43, a value much larger than 4 which has been proposed as a minimum indicator of unidimensionality (*Reeve et al., 2007*). All of these measures strongly suggest that a single factor is sufficient to account for the data on this scale. The factor loadings for a one-factor solution are shown in Table 2. All loadings are equal to or larger than 0.7 indicating a very strong relationship between these items and the latent construct of OHL. In summary, these results consistently suggest that OHL as measured by the HeLD-14 is a unidimensional and well-defined construct.

### CFA results

Based on the EFA results, a one-dimensional model was fit to the confirmatory data. Estimated factor loadings are shown in Table 2. Aside from the $\chi^2$ test, the fit measures indicate evidence for the unidimensional model. Specifically, $\chi^2(77) = 178.9, p < .001$, RMSEA $= .065$, SRMR $= .074$, CFI $= .992$, and TLI $= .991$. In conclusion, the large

**Table 1  Characteristics of the study population and comparisons to HeLD-14 and OHIP-5 sum scores.**

| | Mean (SD) | n (%) | HeLD-14 sum score | OHIP-5 sum score |
|---|---|---|---|---|
| Age ($n=626$) | 43.6 (17.6) | | .29 | −.01 |
| Gender ($n=626$) | | | | |
|   Male | | 196 (31.3) | 48.80 (9.05) | 3.12 (3.38) |
|   Female | | 419 (66.9) | 51.36 (6.34) | 2.91 (3.27) |
|   Non-binary | | 5 (0.8) | 45.20 (9.58) | 3.40 (3.85) |
|   Prefer to self-describe | | 1 (0.2) | 53.00 (0.00) | 4.00 (0.00) |
|   Prefer not to answer | | 5 (0.8) | 49.00 (7.35) | 0.80 (1.30) |
| Education ($n=615$) | | | | |
| Some high school w/out graduating | | 2 (0.3) | 51.00 (4.24) | 8.50 (12.02) |
|   HS graduation or GED | | 46 (7.3) | 47.85 (11.78) | 4.02 (4.13) |
|   Some college/2 yr degree | | 140 (22.4) | 49.43 (8.47) | 3.41 (3.57) |
|   4 yr college graduate | | 197 (31.5) | 51.11 (6.31) | 2.43 (2.53) |
|   More than 4 yr college degree | | 241 (38.5) | 51.10 (6.38) | 2.80 (3.34) |
| Ethnicity ($n=575$) | | | | |
|   Non-Hispanic | | 556 (96.7) | 48.26 (9.51) | 3.79 (3.63) |
|   Hispanic | | 19 (3.3) | 50.67 (6.96) | 2.88 (3.13) |
| Race ($n=631$)[*] | | | | |
|   White | | 567 (89.9) | 51.09 (6.34) | 2.88 (3.14) |
|   Black/African American | | 14 (2.2) | 43.79 (14.94) | 5.07 (3.36) |
|   Asian | | 42 (6.7) | 43.64 (11.66) | 3.57 (4.30) |
|   Native Hawaiian/Pacific Islander | | 2 (0.3) | 42.50 (13.44) | 8.00 (0.00) |
|   American Indian/Alaskan Native | | 5 (0.8) | 46.20 (4.97) | 6.80 (7.66) |
|   Other | | 11 (1.7) | 49.91 (6.07) | 3.46 (2.84) |
| HeLD-14 total score ($n=631$) | 50.5 (7.5) | | —— | —— |
| OHIP-5 Summary Score ($n=630$) | 3.0 (3.3) | | —— | —— |

**Notes.**

[*]Participants were allowed to select more than 1 race category.

Statistically significant results ($\alpha = .05$) using either correlation test (age), Tukey's HSD test (gender, education), and pairwise $t$-tests (ethnicitiy, race). Results that are not statistically significant are not reported.

- Correlation between age and HeLD-14 sum scores: $t(624) = 7.60$, $p < .001$.
- HSD between male and female and HeLD-14 sum scores: $p < .001$.
- HSD between high school graduation or GED *vs.* more than 4 yr college degree on HeLD-14 sum scores: $p = .049$.
- HSD between high school graduation or GED *vs.* 4 yr college degree on OHIP-5 sum scores: $p = .046$.
- White *vs.* not White on HeLD-14 sum scores: $t(66.67) = -3.73$, $p < .001$.
- Asian *vs.* not Asian on HeLD-14 sum scores: $t(43.03) = 4.02$, $p < .001$.
- Black/African American *vs.* not Black/African American: $t(13.57) = -2.37$, $p = .03$.
- Native Hawaiian/Pacific Islander *vs.* not Native Hawaiian/Pacific Islander: $t(627) = -38.37$, $p < .001$.

factor loadings that are consistent across the EFA and CFA provide strong support for a unidimensional model of OHL. Coefficient Alpha for the CFA dataset was 0.92, indicating good reliability (*Nunnally, 1978*).

The EFA and CFA results indicated a unidimensional model allowing the authors to accept the study hypothesis.

**Table 2   Descriptive statistics and factor loadings for the HeLD-14 EFA and CFA models.**

| Item | Average response (IQR) $n = 631$ | EFA loading $n = 316$ | CFA loading $n = 315$ |
|---|---|---|---|
| Are you able to pay attention to your dental or oral health needs? | 3.5 (3-4) | 0.83 | 0.80 |
| Are you able to make time for things that are good for your dental or oral health? | 3.4 (3-4) | 0.76 | 0.74 |
| Are you able to read written dental information (e.g., leaflets given to you) by your dentist? | 3.7 (4-4) | 0.82 | 0.78 |
| Are you able to take family or a friend with you to a dental appointment? | 3.4 (3-4) | 0.69 | 0.71 |
| Are you able to pay to see a dentist? | 3.5 (3-4) | 0.81 | 0.77 |
| Are you able to pay for medication to manage your dental or oral health? | 3.5 (3-4) | 0.80 | 0.81 |
| Do you know how to get to a dentist's appointment? | 3.8 (4-4) | 0.86 | 0.89 |
| Are you able to look for a second opinion about your dental health from a dental health professional? | 3.4 (3-4 ) | 0.79 | 0.77 |
| Are you able to use information from a dentist to make decisions about your dental health? | 3.7 (4-4) | 0.93 | 0.89 |
| Are you able to carry out dental instructions that a dentist gives you? | 3.7 (4-4) | 0.90 | 0.90 |
| Are you able to use advice from a dentist to make decisions about your dental health? | 3.7 (4-4) | 0.91 | 0.93 |
| Are you able to read dental or oral health information brochures left in dental clinics and waiting rooms? | 3.8 (4-4) | 0.88 | 0.81 |
| Are you able to ask someone to go with you to a dental appointment? | 3.6 (3-4) | 0.81 | 0.73 |
| Do you know what to do to get a dentist's appointment? | 3.8 (4-4) | 0.94 | 0.92 |

**Notes.**

IQR, Interquartile range.

Observed responses were on a 5-point Likert scale where 0 = "unable to do", 4 = "no difficulty". Average response and IQR are based on the total ($n = 631$) data set.

## Concurrent validity results

The Pearson correlation between HeLD-14 total scores and OHIP-5 sum scores was $-0.51$, $p < 0.001$, confirming the predictive ability of the instrument.

## Post-hoc association analysis

Due to the discrepancies between the distribution within each group of demographic characteristics, association analyses were run for both the HeLD-14 and OHIP-5 sum scores. Pearson correlations were conducted for age and HeLD-14 sum scores, Tukey's HSD test was used for gender and education, and pairwise t-tests were used for ethnicity and race. Alpha was established at 0.05. Statistically significant results were found for the correlation between HeLD-14 sum scores and age ($t$ (624) $=7.60$, $p < 0.001$), and the HSD between male and female gender ($p < 0.001$). Comparisons of high school graduation/GED and more than a four-year degree with HeLD-14 sum scores, and of high school graduation/GED and a four year college degree with OHIP-5 sum scores both yielded $p = 0.05$. The significance test result for White *vs.* non-White and HeLD-14 sum scores was $t$ (66.67) $= -3.73$, $p = 0.03$.

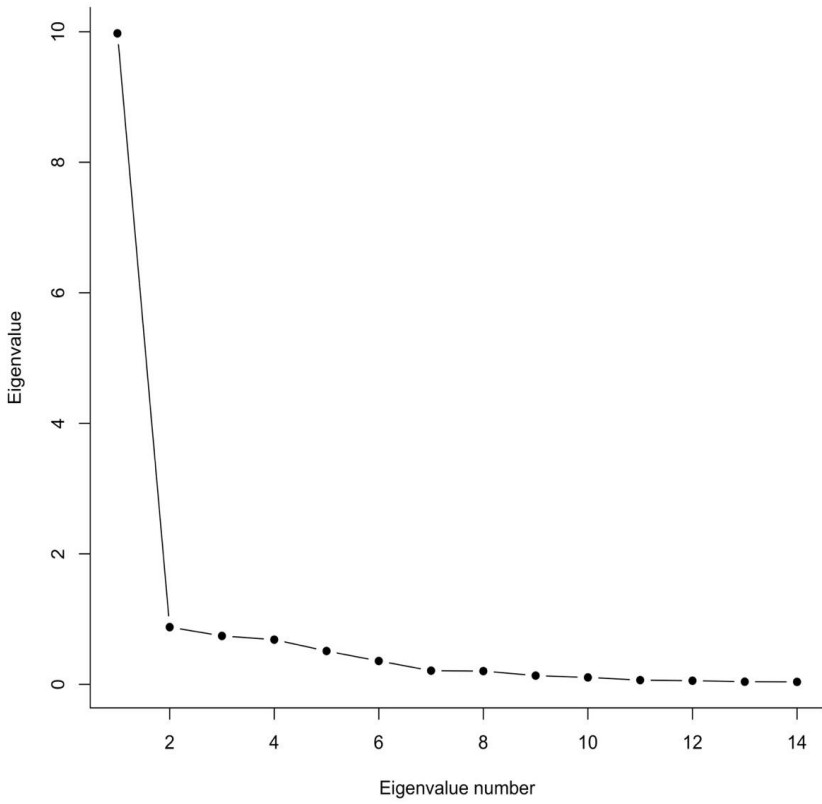

**Figure 1  Scree plot of HeLD-14 EFA results.**

## DISCUSSION

This is the first study to evaluate the psychometric properties of the HeLD-14 in a general US population. The EFA results suggest a one-factor solution and indicate that the latent construct of OHL is a unidimensional construct. Our study is in line with *Jones et al. (2014)* who found a single factor solution for the HeLD-29 in an Australian indigenous population when the eigenvalue greater than one criterion was used to determine the number of factors (*Jones et al., 2014*). No model fit results or further explanation was provided (*Jones et al., 2014*). *Lui et al. (2021)* also reported a single-factor structure of the HeLD-C in the adult Chinese population. A better fit was found in our study compared to the HeLD-C for RMSEA (0.067 *vs* 0.08, respectively) as adequate fit models less than 0.07 are desirable. Similar results indicating adequate model fit were found in both our study and the HeLD-C for SRMR (0.07 *vs* 0.05, respectively), CFI (0.992 *vs* 0.95, respectively), and TLI (0.991 *vs* 0.94, respectively) (*Lui et al., 2021*).

In contrast, the validation of the HeLD-14 in English among an Australian Aboriginal English-speaking population did not report model fit analysis (*Jones et al., 2015*), thus compromising comparisons with our results. The authors conceptualized a seven-dimensional structure, and found that six factors loaded onto the model as expected, but found that two items dispersing onto another single factor (*Jones et al., 2015*). The

Brazilian version of the HeLD-14 administered to 535 elderly adults found the same six dimensions as (*Soares et al., 2022*) using psychometric network analysis, but only after modification of the original hypothesized model (*Jones et al., 2015*). However, adequate fit results were reported ($\chi^2 = 241.01$, $p < 0.001$; CFI $=0.97$; RMSEA $=0.064$) (*Soares et al., 2022*). An exploratory factor analysis of the Thai version (HeLD-Th) among a small group of adults with physical disabilities ($n = 160$) extracted six intercorrelated factors with eigenvalues greater than one. As all communalities were more than 0.4, all seven dimensions were accepted (*Sermsuti-Anuwat & Pongpanich, 2019*).

Discrepancies in findings may be due, in part, to various analytic approaches used to procure results, or lack thereof. *Soares et al. (2022)* used a novel network analysis approach and confirmed the six-factor result reported by *Jones et al. (2015)*. However, it is not clear how this approach identified the number of factors and how it differs from traditional EFA and CFA. Further, the initial analysis did not yield the hypothesized structure, and modifications were made to yield the directed results (*Soares et al., 2022*). *Sermsuti-Anuwat & Pongpanich (2019)* reported six dimensions for the HeLD-Th version, although discrepancies were found in factor loadings. The HeLD-Th reported that the constructs of "Understanding" and "Access" loaded onto the same factor unlike "Communication" and "Utilization" loading onto the same factor in the Brazilian study (*Soares et al., 2022*; *Sermsuti-Anuwat & Pongpanich, 2019*).

While *Lui et al. (2021)* used exploratory structural equation modeling, they limited their analysis to 13 items as one payment-related item was not relevant to the Chinese population (*Nunnally, 1978*). Regardless, their results were similar to this study's and used similar analytic methods. The elimination of one item would not affect the outcome when loading onto a unidimensional model (*Hu & Bentler, 1999*).

The previous validation studies were conducted in diverse populations: lower income Australian Aboriginals, wealthy Brazilians, Thai adults with developmental disabilities, and dental outpatients in a Chinese hospital (*Jones et al., 2014*; *Jones et al., 2015*; *Lui et al., 2021*; *Soares et al., 2022*; *Sermsuti-Anuwat & Pongpanich, 2019*). The majority of participants in this study were White non-Hispanic Americans with high levels of educational attainment. While it is beneficial to test an instrument in diverse cultures and populations, more studies are required to further explore the dimensionality of the HeLD. It is not uncommon for researchers to report different dimensional structures of new instruments exploring latent variables, as with the related but distinct concept of OHRQoL.

Findings from studies of OHRQoL dimensionality using the OHIP instrument shed some light on the interpretation of oral health literacy's dimensionality. The OHIP, like the original HeLD instrument, started with a seven-dimensional structure (*Slade & Spender, 1994*). Later a short instrument, the OHIP-14 followed (*Slade, 1997*). OHIP-14 and HeLD-14 have the same number of initially proposed domains/dimensions as well as items for each dimension. When OHRQoL was initially researched, several dimensional studies were performed, in line with our current work with the HeLD (*Mumcu, Hayran & Ozalp, 2007*; *Segu et al., 2005*). These early studies all consistently rejected the seven OHRQoL domain structure until finally a four-dimensional structure was identified and confirmed (*John, 2021*). The four-dimensional structure, a concept with world-wide applicability (*Ingleshwar*

& *John, 2023*), has also, as well as OHL, a strong general factor (*John et al., 2014*). This could also be shown for the shortest oral health impact profile, the OHIP-5 (*John et al., 2006*) an instrument that performs consistently across different settings (*Simancas-Pallares et al., 2020*; *Larsson et al., 2014*), and is recommended for OHRQoL assessment across multiple settings and oral conditions (*John, 2022*; *John et al., 2021*; *John et al., 2022*).

Based on the similarities between OHRQoL and OHL, it can be expected that a simple unidimensional structure is a reasonable fit for OHL and based on such findings further reduction in the number of items can be expected to increase practicability of all OHL instruments applications across settings.

Results of this study confirmed the positive relationship between OHL and OHRQoL such that greater oral health literacy is correlated with better OHRQoL.

Association analyses of demographic characteristics with both HeLD-14 and OHIP-5 sum scores found that HeLD-14 scores increased with age. This result is consistent with one previous study reporting similar results (*Lui et al., 2021*), but inconsistent with previous HeLD-29 and HeLD-14 results where sum scores decreased with increasing age (*Jones et al., 2015*; *John et al., 2022*). Women in the current study had significantly greater HeLD-14 scores compared to men which is consistent with two other studies although differences did not reach statistical significance or comparisons were not reported (*Lui et al., 2021*; *Sermsuti-Anuwat & Pongpanich, 2019*). However, *Jones et al. (2015)* reported that males had higher HeLD-29 and HeLD-14 sum scores compared to women ($p < 0.05$) (*Jones et al., 2015*). This study found that those graduating from high school/GED had lower HeLD-14 scores than those with more than a four-year college degree ($p < 0.05$). These results are consistent with those found for both the HeLD-29 and HeLD-14 (*Jones et al., 2015*). However, both *Bado et al. (2020)* and *Sermsuti-Anuwat & Pongpanich (2019)* found no differences in HeLD-14 scores by educational level. White participants had higher HeLD-14 scores than non-Whites in this study. Of published studies using the HeLD-14, only one reported race (*Bado et al., 2020*) and found no statistically significant differences. The only statistically significant finding between OHIP-5 sum scores and demographic characteristics in the current study was between participants with educational attainment of high school graduation/GED and a four-year college degree ($p < 0.05$) indicating improved OHRQoL with increasing education. This is consistent with similar studies where more years of education were also associated with improved OHRQoL (*Husain & Tatengkeng, 2017*; *Wide & Hakeberg, 2018*). However, other studies have found no statistical significance association between education and OHIP scores (*Bado et al., 2020*; *Ikebe et al., 2012*), or decreased OHRQoL with increasing educational attainment (*Collins et al., 2019*; *Fernanes et al., 2006*). These differences may reflect variation in educational systems throughout the world, or differences in survey categories used to measure educational attainment.

## Strengths and limitations

Our study has several strengths. Our sample size exceeded the minimum to investigate a unidimensional structure with high factor loadings. Samples sizes as low as 100 are sufficient to accurately recover population parameters (*MacCallum et al., 1999*). Model fit indices all pointed into the same direction of adequate model fit. Only the chi-square statistic did

not support model fit, but this statistic is known to be sensitive to sample size with larger samples regularly indicating significantly poor fit where there may only be a trivial model misfit (*Babyak & Green, 2010*). A further strength is our two-step approach to investigating dimensionality. Both methods agreed on the conclusion that unidimensionality is present. Finally, the number of missing data was small.

Limitations of our study include a lack of information about participant non-response because our study participants self-selected themselves for participation. That is, our sample is a convenience sample. Likely, the study design characteristic does not influence results because correlations among HeLD-14 items should be reasonably stable across populations.

## CONCLUSIONS

Our results indicate that the construct of OHL is likely unidimensional and that the HeLD-14 is an appropriate instrument to measure OHL in a general US English speaking population. Additional studies are needed in diverse populations to further explore the dimensionality of OHL.

### Funding
The authors received no funding for this work.

### Competing Interests
The authors declare there are no competing interests.

### Author Contributions
- Priscilla Flynn conceived and designed the experiments, performed the experiments, prepared figures and/or tables, authored or reviewed drafts of the article, and approved the final draft.
- Aparna Ingleshwar performed the experiments, analyzed the data, prepared figures and/or tables, authored or reviewed drafts of the article, and approved the final draft.
- Xing Chen analyzed the data, prepared figures and/or tables, authored or reviewed drafts of the article, and approved the final draft.
- Leah Feuerstahler analyzed the data, prepared figures and/or tables, authored or reviewed drafts of the article, and approved the final draft.
- Yvette Reibel performed the experiments, authored or reviewed drafts of the article, and approved the final draft.
- Mike T. John conceived and designed the experiments, authored or reviewed drafts of the article, and approved the final draft.

### Human Ethics
The following information was supplied relating to ethical approvals (i.e., approving body and any reference numbers):

Ethical approval. The study was approved by the University of Minnesota Institutional Review Board.

## Data Availability

The raw data are available in the Supplementary File.

## Supplemental Information

Supplemental information for this article can be found online at http://dx.doi.org/10.7717/peerj.16106#supplemental-information.

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
