# Peer review of "Validation of the HeLD-14 functional oral health literacy instrument in a general population"

_PeerJ, doi:10.7717/peerj.16106_

## Round 0.1 · original submission · Major Revisions

I have carefully reviewed your manuscript titled "Validation of the HeLD-14 functional oral health literacy instrument in a general population" and would like to acknowledge the significant effort put into the research. However, before proceeding with its potential publication, I must request major revisions. Two crucial areas that need improvement are the Sample Size Calculation and Ethical Approval. The reviewer highlights the need for a clear rationale and robust statistical methods for sample size determination, along with a power analysis. Additionally, the manuscript lacks comprehensive information on the ethical procedures, such as the ethics committee's name, approval number, and informed consent process for participants.

Reviewer 1 ·

Basic reporting

This manuscript is well written and referenced. I applaud the investigators for pursuing this inquiry. The HeLD-14 is a promising instrument for measuring oral health literacy addressing weaknesses in other current OHL measures. This study is quite welcome for the field. If word limitations allow, it might be helpful for those not familiar with the other OHL measures why studying the HeLD-14 in the US population represents such an important, and potentially impactful pivot in OHL studies. For example, how the tool presents dimensions of OHL that could be the target of interventions and improved. Current measures that use word recognition or OHQoL measures do not offer that actionability.

Experimental design

This research is within the scope of the journal.

The study population may reflect an unintentional selection bias. However, the findings demonstrate the tool’s utility as a unidimensional, reliable and valid instrument to measure OHL. The methods are described well, with sufficient detail. The tables are well labeled and easy to understand.

Validity of the findings

The conclusions are sound. I hope the authors, or others, will continue to investigate the HeLD-14’s utility in other populations.

Reviewer 2 ·

Basic reporting

I would like to thank the authors for the opportunity of reviewing this article. The oral healthy literacy is a thematic of great importance the evaluation of methods to measure the OHL are needed. The article is well-written and seems adequate for acceptance after some minor adjusts.
Introduction
1- Please, add the study hypothesis at the end of the introduction section.
2- Since the HeLD-14 was compared to the OHIP-5, I think that the last must be introduced and discussed in the section.
Materials and methods
1- Criteria related to the health of the participants should be added for eligibility, such as vision, hearing and motor skills, as these could influence the responses referring to the 7 domains.
Discussion
1- There are some formatting mistakes in the first paragraph of this section, as letter size and some characters. Please review the entire article.
2- Please state if the added hypothesis were accepted or rejected.

Experimental design

Materials and methods
1- Criteria related to the health of the participants should be added for eligibility, such as vision, hearing and motor skills, as these could influence the responses referring to the 7 domains.

Validity of the findings

The findings are properly reported.

Additional comments

No additional comments

Reviewer 3 ·

Basic reporting

The article “Validation of the HeLD-14 functional oral health literacy instrument in a general population” has an interesting topic. The aim is to validate the HeLD-14 questionary to evaluate oral health literacy (OHL) in an English-native speaker population. Some adjustments could be performed.

Experimental design

2.1. Considering as an adaptation of a questionary that was firstly validated in a 400 indigenous Australians population, was there any text modification necessary? If yes, please report which modification and how it was performed.
2.2. Was it performed a sample size calculation for the minimal amount of participants? If not, a power analysis could be interesting.
2.3. I understand the aim of the authors to validate the questionary comparing to an short version of an oral health-related quality of life instrument. However, the references that the authors mention do not support what was performed. Both of the studies used the OHIP-14 as a comparison, and one of them did not compared with the HeLD-14 questionary. Based on that, why not to use OHIP-14 or even the HeLD-29 instrument to validate its short and adapted version?
2.4. What was the statistical software used for the analysis?
2.5. The ethical approvement should be in the beginning of the materials and methods section, not in the analysis, since its expected to be performed before the study conduction.
2.6. Considering the discrepancy among the sample characteristics, the authors could discuss how this affected the results. Therefore, I expected and suggest an association analysis among the descriptive data collected and both questionary results.
2.7. The discussion section is well-argumentative. Please consider removing the topics both to the result and discussion sections.

Validity of the findings

3.1. The findings are valid to an specific native English speaker population. An association analysis among the sample characteristic is recommended in order to understand the specific aspects that are validated through the study.

---

## Round 0.2 · accepted · Accept

Dear Priscilla Flynn,

I am pleased to confirm that all reviewers' comments have been successfully addressed in the revised version of your manuscript titled "Validation of the HeLD-14 Functional Oral Health Literacy Instrument." Your diligent revisions are greatly appreciated.

I have personally reviewed the revisions and am satisfied with the manuscript's current state.

Considering the thorough revisions made, I believe the manuscript is now ready for publication in PeerJ.

Kind regards,
Dr. Tribst JPM

Reviewer 1 ·

Basic reporting

No comment.

Experimental design

no comment

Validity of the findings

no comment.

Additional comments

The revisions and additional analyses are noted and strengthen the paper.

Reviewer 2 ·

Basic reporting

No comment

Experimental design

No comment

Validity of the findings

No comment

Additional comments

The study was properly reviewed and is now suitable for acceptance.

Reviewer 3 ·

Basic reporting

The authors improved the pointed issues. Therefore, I recommend the acceptance.

Experimental design

No comment.

Validity of the findings

No comment.